# What Data Difficulty Metrics Should We Measure for Tabular Deep Learning?

## Abstract

The notion of data difficulty has garnered attention in the machine learning community due to its wide-ranging applications, from noisy label detection to data debugging and pruning. Yet with many competing definitions, researchers and practitioners have often selected difficulty metrics in an ad hoc manner. Further, systematic evaluations have been limited to vision settings, and tabular DL presents unique challenges. To aid principled metric selection in tabular deep learning, we conduct a comprehensive empirical study of existing metrics, including logit-based, gradient-based, ensemble, valuation, and influence methods. By collecting difficulty scores across diverse model architectures, tasks, and epochs, we assemble a large-scale dataset for statistical analysis. We ask and answer the following questions: (1) How many orthogonal factors comprise data difficulty? (2) How many metrics and random seeds are needed to rank difficulty robustly? (3) Which metrics are most effective for noisy label detection? (4) Is the factor structure stable across subgroup splits? (5) How are early-training and late-training difficulty different? Our results contradict both the view that difficulty metrics are neither redundant nor hyper-specialized. Instead, we identify three consistent factors: label-aware difficulty, confidence, and influence/valuation. We show that measuring a computationally inexpensive exemplar of each factor captures most interpretable information, and that the three factors are strongly predictive of noisy labels. We further observe that confidence is more prominent in test data, whereas influence/valuation is more important in train data. Rank stability analysis shows that combining just two metrics, each measured over two random seeds, yields rankings that correlate strongly with the ground truth. Finally, we contribute an open-source Python library that streamlines the measurement of difficulty metrics from model snapshots.

## 1 Introduction

The notion of data difficulty Kwok et al. (2024); Zhu et al. (2024); Meding et al. (2022) has garnered attention for data-centric and explainable machine learning. Difficulty metrics capture how hard an example is for models to learn. Examples of difficulty scores include: high loss Jiang et al. (2021), frequent forgetting Toneva et al. (2019), late learning Baldock et al. (2021), low confidence Qendro et al. (2021), and ensemble disagreement Carlini et al. (2019). The related attribution metrics of valuation and influence instead quantify how much an example contributes to model performance when included or removed from the train set Ghorbani & Zou (2019); Wang & Jia (2023). The two clusters of metrics are thought to be tightly related Paul et al. (2021). Together, these metrics have a wide range of applications, from noisy-label detection Pleiss et al. (2020), curriculum learning Soviany et al. (2022), and subgroup discovery Sagadeeva & Boehm (2021) to dataset distillation Shadin & Zhang (2025), data markets Tian et al. (2022), interpretability Koh & Liang (2017), and data unlearning Li et al. (2024). For practitioners, they offer actionable tools for model debugging and data mining; for researchers, they provide signals for designing data-debugging methods.

**Tabular DL challenges.** Most systematic comparisons of difficulty metrics have been carried out on classic vision benchmarks, where ConvNet variants dominate and near-perfect performance causes metrics to converge Kwok et al. (2024); Carlini et al. (2019); Paul et al. (2021); Meding et al. (2022). Tabular deep learning (TDL) presents distinct challenges. Diverse architectures with

different inductive biases—such as MLPs, ResNets, and transformers—perform well on different datasets Grinsztajn et al. (2022); Tschalzev et al. (2024). Furthermore, unlike classic vision benchmarks, TDL includes many datasets where all deep neural networks underperform Grinsztajn et al. (2022), which may amplify disagreements between metrics. These differences mean that difficulty and valuation signals may not align in TDL as they do in vision, necessitating a dedicated study. Further, compared to vision, in TDL, model re-training costs are cheaper, due to smaller inputs, while data augmentation is less effective, due to rotational non-invariance, which means that applications and scientific experiments have a different tradeoff space that should emphasize model re-training.

**Limitations of prior work.**   Existing comparisons often serve as vehicles to propose new difficulty metrics Baldock et al. (2021); Agarwal et al. (2022). Independent comparisons such as Kwok et al. (2024) and Carlini et al. (2019) are restricted to a few easy vision benchmarks, ConvNet-style architectures, a limited number of random seeds, and the use of PCA for factor analysis Floyd et al. (2009) — all design decisions that inflate spurious correlations and emphasize a single primary factor. Consequently, practitioners lack principled guidance on which metrics to measure in TDL.

**Goal of this study.**   We aim to build an empirical model of difficulty metrics for TDL that is both robust and actionable. For practitioners, we provide cost-effective recipes to measure interpretable factors, mine challenging data slices, and detect noisy labels under various resource constraints. For researchers, we propose a multi-factor model of difficulty and attribution for more principled development of data-debugging methods, to help progress beyond oversimplistic single-factor models and the overspecificity of previously proposed metric-downstream task pairings.

**Study overview.**   We analyze eleven representative metrics (Table 1) across thousands of independently trained models on multiple tabular datasets (Table 2) and architectures. We log both train and test difficulty, track metrics across epochs, and various random seeds. Using modern factor analysis, we extract the orthogonal components of the studied metrics. We confirm that the extracted factors are meaningful in the train and test sets via data visualizations and subgroup analysis. We then assess robustness across subgroups (e.g., model architectures, datasets with high or low #feats, etc.), cost-efficiency, and utility for noisy-label detection. Specifically, we ask: (RQ1) What orthogonal factors underlie difficulty and attribution, and how do they differ between train and test splits? (RQ2) How many metrics and seeds suffice for robust ranking of data by their difficulty? (RQ3) Which combination of metrics is most effective for the downstream task of noisy-label detection? (RQ4) Is the factor model stable across subgroup splits? (RQ5) Is early-training difficulty distinct from late-training difficulty?

**Preview of findings.**   We find that many metrics are moderately correlated to each other, but that they are not reducible to a single linear component. Instead, three consistent orthogonal factors emerge: a core *difficulty* factor (**Dif**), which is split into a label-aware (**Dif-gt**) and a label-free confidence factor (**Dif-conf**); and an *influence/valuation* factor (**Inf/Val**), the latter of which is only present in the train set. Extracting any additional factors produces a spurious factor that explains a single variable. These factors remain stable across datasets, architectures, and meta-features. Consequently, measuring just three high-quality exemplars of each factor (e.g., AUM, confidence, kNN-Shapley) is sufficient and effective for interpretability and noisy label detection. We also discover that robust ranking requires only *two seeds* and *two epoch-wise metrics* per factor, and that early epoch measurements are less valuable for understanding model behavior at convergence. These observations help prune unnecessary measurements for practitioners.

**Contributions.**   1) A comprehensive empirical analysis of data difficulty in TDL, spanning diverse metrics, datasets, and architectures. 2) A three-factor model of difficulty—that distinguishes between ground-truth-based (**Dif-gt**), confidence-based (**Dif-conf**), and valuation-based (**Inf/Val**) metrics—and an empirical analysis of its properties (**RQ1**, **RQ4**, **RQ5**). 3) An analysis of the proposed three-factor model for common downstream applications of difficulty ranking and noisy label detection (**RQ2** and **RQ3**). 4) Cost-efficient recommendations for practitioners to measure difficulty robustly for different goals and resource constraints. 5) An open-source library[1] that streamlines evaluating various metrics from model snapshots (details in Appendix 5.4).

---

[1]`https://anonymous.4open.science/r/easydatdif-3F71`

| Name | Definition | Reference |
|------|-----------|-----------|
| Loss | The sample-wise cross entropy loss. | |
| Confidence | The highest value in the softmax probabilities. | |
| Area Under the Margin (AUM) | The logit value for the ground truth label minus the highest logit value for any other label. | Agarwal et al. (2022) |
| Forgetting Events | The number of times the model went from correctly classifying the data to incorrectly classifying it. | Toneva et al. (2019) |
| Learned Epoch | The earliest epoch at which the model correctly classifies the data and does not proceed to forget it. | Baldock et al. (2021) |
| Ensemble Agreement | The Shannon entropy of an ensemble's logits concatenated and normalized into a probability. | Carlini et al. (2019) |
| Gradient Normed (GRAND) | The L2 norm of model gradient. Approximated via the final layer. Originally measured once early. | Paul et al. (2021) |
| Variance of Gradients (VOG) | The variance of gradient over training averaged across channels. We use the fc layer instead of pixels. | Agarwal et al. (2022) |
| Gradient Similarity | The cosine similarity between the sample-wise final layer gradient and the mean gradient. | Dhaliwal & Shintre (2018) |
| TracIn | Sum of learning-rate scaled dot product of test sample and train sample's final layer gradient. | Pruthi et al. (2020) |
| kNN-Shapley | Data Shapley value assuming a kNN classifier over the final layer activations. ($k = 5$) | Wang et al. (2024) |

Table 1: The data difficulty metrics included in our study.

## 2 METHODS

**Design requirements.** We followed the following high-level requirements to design this study. Our analysis must be statistically robust by controlling spurious correlations across seeds, datasets, and metrics. The collection of metrics should be mechanically diverse and avoid trivial factors and multicollinearity. We include attribution (valuation, influence) metrics as a hedge against construct underrepresentation, as our work is not the first time that what were thought to be distinct concepts are highly correlated to each other Carlini et al. (2019); Kwok et al. (2024). The datasets should be sufficiently diverse in terms of meta-features, hardness of the task, and characteristics that favor different architectures. We include MLP, ResNet, and Transformer-style architectures, which are all SOTA in different datasets.

### 2.1 DATA COLLECTION

We collected data difficulty measurements by training independent random initializations of TDL models over model-architecture combinations. A "task" is an OpenML dataset with an associated target variable. For each task, 20% of the data was randomly designated as the test set, which is excluded from model training and is used for influence and valuation metrics. Additionally, 10% of the train and test splits, respectively, had their labels randomly corrupted. This is used for determining which metrics are useful for noisy label detection. Label corruption also introduces a class of hard and noisy samples which has potentially different properties from hard and clean samples. The result is a long table with the following attributes: TASK, ARCHITECTURE, RANDOM SEED, EPOCH, METRIC, VALUE, IS_TRAIN, and LABEL_CORRUPT. We recorded the measurement of 11 data difficulty metrics over all tasks, model architectures, random seeds, and epochs. To ensure independence, each random seed was used to measure only one metric. Finally, we aggregated and normalized the measurements. Specifically, for each task-architecture-metric tuple, values were averaged across seeds, then quantile-normalized into the standard normal distribution. This yielded a dataset where each task, architecture, and metric is associated with one normalized difficulty value.

**Metrics.** We chose a representative subset of data difficulty and related metrics without exhaustive enumeration. We included distinct mechanistic explanations, including: simple baselines; those derived from logits, gradients, accuracy history, or ensemble; influence and valuation proxies. Measures of difficulty for an entire task are outside the scope of this paper Lorena et al. (2019); Ethayarajh et al. (2022); Boopathy et al. (2023); Rivolli et al. (2022); Bilalli et al. (2017). We also excluded metrics that are prohibitively expensive to measure thousands of times, such as combinatorial valuations Ghorbani & Zou (2019); Li & Yu (2024), second-order methods Koh & Liang

| Dataset Name | OpenML ID | $n$ | #feats | Best Architecture | Mean Val Acc. |
|---|---|---|---|---|---|
| audiology | 7 | 226 | 70 | STG | 0.772 |
| lymph | 10 | 148 | 19 | XGBoost | 0.877 |
| mfeat-fourier | 14 | 2000 | 77 | SVM | 0.831 |
| colic | 27 | 368 | 27 | CatBoost | 0.842 |
| credit-g | 31 | 1000 | 21 | ResNet | 0.749 |
| elevators | 216 | 16599 | 19 | TabNet | 0.675 |
| monks-problems-2 | 334 | 601 | 7 | SAINT | 1.000 |
| balance-scale | 997 | 625 | 5 | TabPFN | 0.984 |
| cnae-9 | 1468 | 1080 | 857 | TabTransformer | 0.670 |

Table 2: Datasets included in our study. We used the default task (target attribute) of each dataset in OpenML. The best architectures were found by McElfresh et al. (2023). Target labels that appear in only one row were combined. Continuous targets were converted into four-quantile classification problems. Mean validation accuracy across the included model architectures before label corruption across three-fold cross-validation.

(2017), and inference depth Baldock et al. (2021). We additionally excluded training-free or model-agnostic metrics such as instance hardness Torquette et al. (2022); Smith et al. (2014) and $\mathcal{V}$-usable information Ethayarajh et al. (2022). Finally, we excluded some trivial variants of other metrics to avoid spurious factors, such as (top-$k$) accuracy, EL2N Paul et al. (2021), and other methods of aggregating epoch-wise metrics (e.g., variance). Table 1 shows the definition of the included metrics.

**Datasets & Tasks.** We selected challenging tabular classification benchmarks from the TabZilla suite McElfresh et al. (2023) with varying dataset sizes, feature counts, and best-performing architectures. It intentionally includes datasets where non-neural nets perform well. We selected the datasets before observing any measurements to help mitigate biased selection. Table 2 lists the datasets included in our study.

**Model Architectures.** We included four TDL architectures with distinct inductive biases: MLP, ResNet Gorishniy et al. (2021), TabTransformer Huang et al. (2020), and SAINT Somepalli et al. (2021). All four have been shown to perform competitively on at least some tabular benchmarks McElfresh et al. (2023); Tschalzev et al. (2024). All architectures use categorical column embeddings Guo & Berkhahn (2016).

## 2.2 STATISTICAL ANALYSIS

**Summary of Analyses.** We performed various statistical analyses to answer the research questions defined in Section 1. To answer **RQ1**, we computed correlations over the aggregated data and performed factor analysis. To answer **RQ2**, we ranked data points using subsets of seeds or metrics, then computed the Spearman's correlation between the noisy rankings (computed from a subset of seeds or metrics), and "ground-truth" rankings (computed from all available data). Ranking data by difficulty with more than one difficulty metric is done by projecting each sample's difficulty measurements onto the basis vector that defines **Dif-gt**. To answer **RQ3**, we examined the correlations between the difficulty metrics with the LABEL_CORRUPT variable, and fitted linear models to predict LABEL_CORRUPT. To answer **RQ4**, we stratified data by architecture, dataset size, feature count, training phase, and the overall hardness of the task. We then repeated correlation and factor analysis within each subgroup.

**Correlation Computation.** We produced the correlation matrix by computing Pearson's R values using mean-aggregated and normalized values. We used importance weighting to ensure that each dataset with a different number of samples contributed equally. Statistically insignificant correlations ($p > 0.05$ via Student's t-test) were set to zero to remove spurious correlations.

**Factor Analysis.** We performed factor analysis, on correlation matrices, using standard techniques Kline (2014) that minimize spurious factors and overinflated primary factor. We conducted principal factor analysis (PFA) using squared multiple correlation (SMC) values on the diagonal to account for imperfect autocorrelations, and rotated the factors for interpretability. We found that the Kaiser criterion consistently under-extracted meaningful dimensions. We chose the number of factors to extract by taking the maximum number that prevents single-variable factors. A factor is a

single-variable factor for our purposes if it has a higher absolute loading compared to other factors on only one variable. Using this criterion, we chose three factors from train splits and two from test splits. Note that test splits have 9 variables in total as opposed to 11, since influence function or data valuation are not well-defined over test data. We additionally tested the robustness of each factor via its stability across subgroups.

**Rank Stability Analysis.** For rank stability analysis, we established the "ground truth" rankings of data points per task-architecture pair by aggregating across all seeds (for #seeds-based analysis) or by projecting data onto the **Dif** factor using metrics that loaded heavily on it (for #metrics-based analysis). We then sampled subsets of seeds (#seeds from 1 to 10) or metrics (#metrics from 1 to 8) up to 50 times without replacement, produced noisy rankings, and computed Spearman correlations of the noisy rankings against the ground truth. This process was repeated for every task-architecture pair, and then the data across tasks and architectures were pooled together.

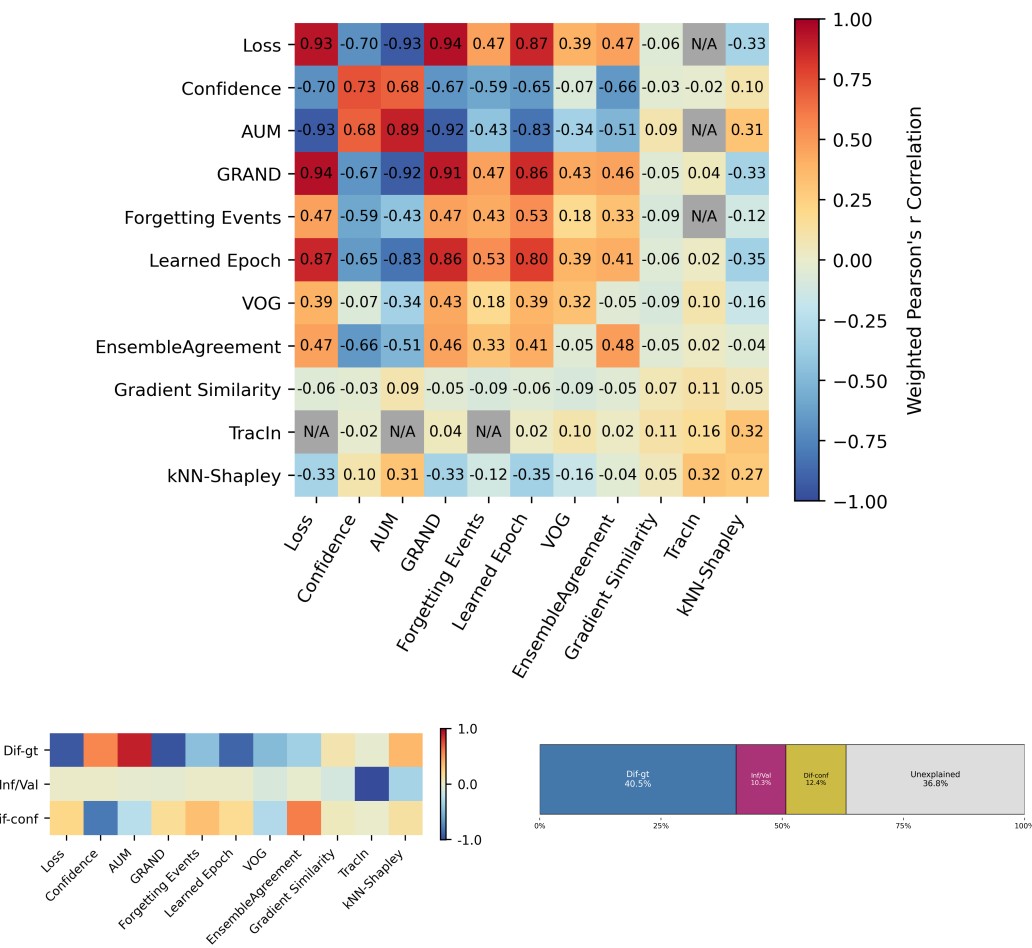

Figure 1: **(Top)** Correlation matrix between the measured data difficulty metrics across all tasks, architectures, random seeds, and train/test split. Pearson's R values with importance weighting are used to ensure each task has the same weight as any other task with a different number of samples. **(Left)** Loadings of the three significant factors extracted from factor analysis on the correlation matrix. **(Right)** PVE of the three factors. For most metrics, higher values correspond to more difficult or more influential data, except confidence, AUM, and gradient similarity, for which the direction is flipped. Gray 'N/A' cells are statistically insignificant ($p > 0.05$).

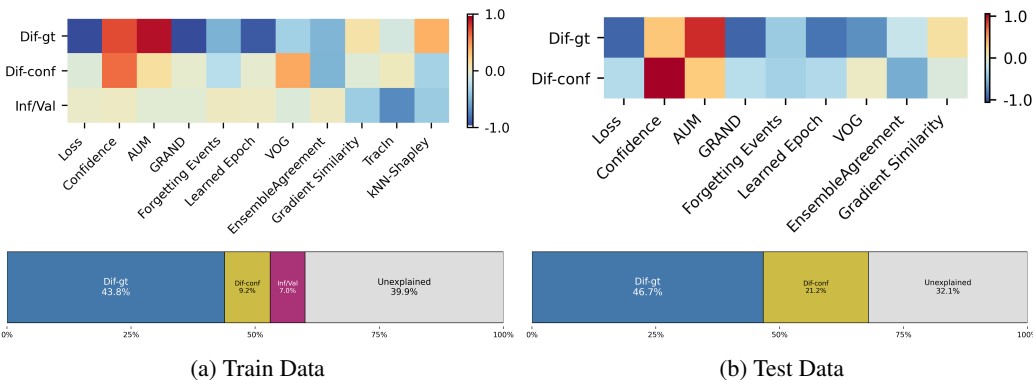

(a) Train Data                        (b) Test Data

Figure 2: The significant factors extracted from performing factor analysis on the correlation matrices, with data split by train and test sets. The number of factors to extract for each split is determined by incrementally increasing the number of factors while ensuring that there is no single-variable factor. A single-variable factor has a higher absolute loading on only a single variable compared to other factors. (Figure 11, Appendix 5.5.) **(Top)** The loadings of the top-3 factors of each split onto difficulty metrics. **(Bottom)** The proportion of total variance explained (PVE).

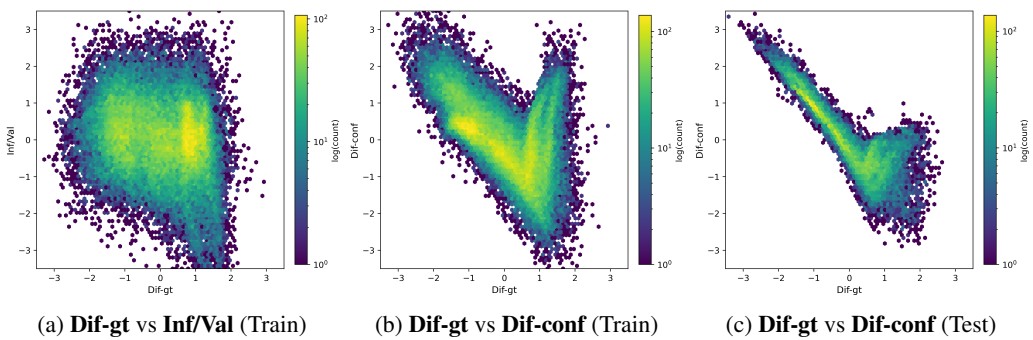

(a) **Dif-gt** vs **Inf/Val** (Train)   (b) **Dif-gt** vs **Dif-conf** (Train)   (c) **Dif-gt** vs **Dif-conf** (Test)

Figure 3: Scatter plot of the joint distribution of the three factors. Higher (positive) **Dif-gt**, **Inf/Val**, and **Dif-conf** values correspond to more difficult, influential/valuable, and confidently predictable data, respectively, and vice versa for lower (negative) values. The colors denote the # of data points across all task-architecture pairs that have a specific joint value for two of the three factors.

## 3 RESULTS

We now present the results, organized by the main RQs. Throughout, we describe the absolute value of correlation as follows: $<0.3$ = weak, 0.3–0.6 = moderate, 0.6–0.9 = strong, $>0.9$ = very strong.

### 3.1 HOW MANY ORTHOGONAL FACTORS COMPRISE DATA DIFFICULTY AND RELATED CONCEPTS? HOW IS TRAIN AND TEST DIFFICULTY DIFFERENT? (**RQ1**)

To answer **RQ1**, we computed correlations between metrics (Figure 1) as well as IS_TRAIN (Figure 6), and performed factor analysis (Figure 2).

We observe moderate to strong correlations between most difficulty metrics. The magnitude of correlations is weaker than those reported in Kwok et al. (2024), likely because we removed spurious correlations by using independent random seeds. There are even negatively correlated pairs of metrics that were positively correlated in Kwok et al. (2024) and Carlini et al. (2019). Gradient similarity, TracIn, and kNN-Shapley are more strongly correlated with each other than with other metrics, which we attribute to their shared reliance on approximating the leave-one-out (LOO) valuation metric, or aggregating the dot product of sample-wise and mean gradient.

By incrementally increasing the number of extracted factors until single-variable factors appeared, we identify three multi-variable factors of **Dif-gt**, **Dif-conf**, and **Inf/Val**. **Dif-gt** primarily loads onto loss, AUM, GRAND, forgetting events, and learned epoch; **Dif-conf** loads onto confidence, ensemble agreement, and VOG; and **Inf/Val** loads onto TracIn, kNN-Shapley, and gradient similarity. We then repeated the same factor analysis separately for train and test splits (Figure 2). In both cases, **Dif-gt** explains the largest variance with a Proportion of Variance Explained (PVE) of approximately 45%, which is markedly lower than the 85% reported in Kwok et al. (2024). Further, we also see that the train set data is slightly easier than the test set data, which is not surprising (Figure 6).

We also visualize the joint distribution of the three factors in the train and test splits, respectively (Figure 3). We see that the joint distribution between **Dif-gt** and **Inf/Val** forms a single cluster. On the other hand, **Dif-gt** and **Dif-conf** form a distinct V-shape, reminiscent of Baldock et al. (2021)'s notions of "validation difficulty" and "train difficulty." The V-shape cannot be fully explained by the presence of noisy labels, which are only 10% of all data; a large subset of clean data is simultaneously difficult and causes overconfidence.

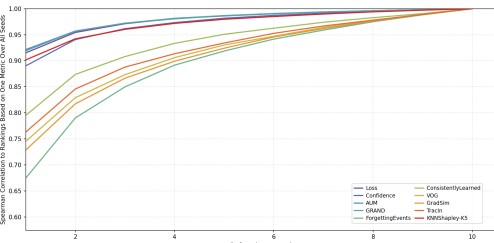

Figure 4: Robustness of ranking data by a single difficulty metric with only a subset of seeds. Measured in terms of Spearman's correlation with the ranking from all seeds.

Figure 5: Robustness of ranking data by **Dif** with only a random subset of metrics. Measured in terms of Spearman's correlation with the ranking from all metrics.

## 3.2 How many metrics should we measure to find the most difficult data robustly? (**RQ2**)

To answer **RQ2**, we compared the ranking of samples by their difficulty from subsets of seeds or metrics against "ground truth" rankings (all seeds and/or metrics) using Spearman's correlation. There are steep diminishing returns in terms of both the number of seeds and metrics. Figure 4 shows that the five epoch-wise metrics of loss, confidence, AUM, GRAND, and kNN-Shapley are significantly more robust with few seeds than run-wise metrics. The most to least robust metrics are, in order: AUM, GRAND, loss, kNN-Shapley, confidence, learned epoch, TracIn, gradient similarity, VOG, and forgetting events. Figure 5 shows that two random difficulty metrics suffice for the median random subset to be very strongly correlated to the ground truth, and three metrics suffice for the mean random subset, when ranking data by **Dif-gt**. We conjecture that similar results hold for the **Dif-conf** and **Inf/Val** factors.

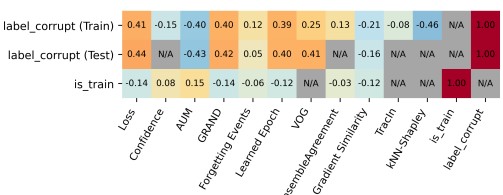

Figure 6: Correlation between difficulty metrics and auxiliary variables IS_TRAIN and LABEL_CORRUPT.

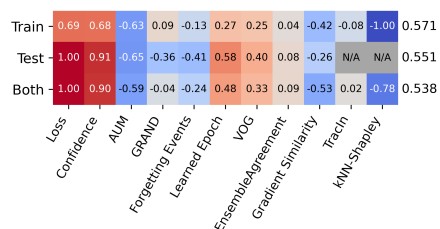

Figure 7: Coefficients that produce the optimal linear combination of metrics to predict label corruption, and the strength of prediction.

### 3.3 WHAT METRICS SHOULD BE USED FOR NOISY LABEL DETECTION? (RQ3)

Figure 6 shows the correlation between the difficulty metrics and the LABEL_CORRUPT auxiliary variable in train and test sets. In train sets, kNN-Shapley is the strongest predictor of label corruption, followed by loss, AUM, GRAND, and learned epoch in a four-way tie. In test sets, loss and AUM outperform all other metrics. Yet, the correlations in the 0.4-0.5 range are only moderate.

We can better predict label corruption by forming a linear combination of the difficulty metrics, with the optimal coefficients given in Figure 7. This raises the strength of correlation to 0.54-0.58. Notably, the variables with the highest coefficients correspond to the three-factor model. For train data, we predict label corruption with low kNN-Shapley, low AUM, and high confidence, which corresponds to data that has low valuation (**Inf/Val**), high difficulty (**Dif-gt**), and overconfidence (**Dif-conf**). For test data, the role of confidence (**Dif-conf**) rises as **Inf/Val** is absent.

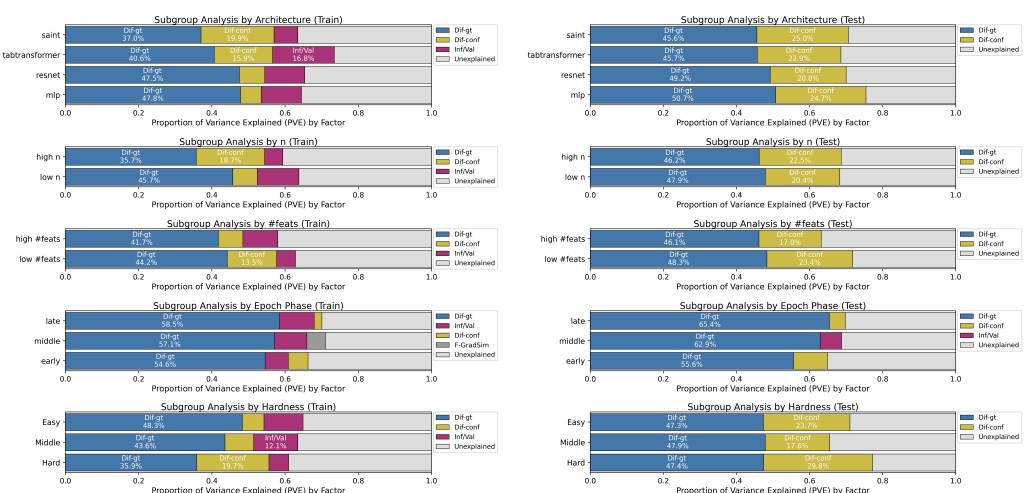

Figure 8: Summary of various subgroup analyses, where we stratify the available data, then perform factor analysis on each subgroup. We extract an interpretable factor name and compute the PVE for each factor. **(Left)** Train sets. **(Right)** Test sets. **(Top to Bottom)** Subgroups by model architecture, number of rows, number of features, training phase, and dataset hardness (mean val acc.). Epoch-wise subgroup analysis only includes epoch-wise metrics.

### 3.4 IS THE THREE-FACTOR MODEL ROBUST TO SUBGROUP SPLITS?

To test **RQ4**, we stratified the full data into different subgroups, then performed correlation and factor analysis in each. To summarize the results, we identified the interpretable name of each factor based on the metrics that it has the highest absolute loadings onto, and computed their PVEs (Figure 8). We extracted 3 factors from train splits and 2 factors from test splits, as above.

In train sets, **Dif-gt** and **Inf/Val** factors are consistently extracted; **Dif-Conf** is extracted in the majority of subgroups. In test sets, **Dif-gt** and **Dif-Conf** are extracted in all subgroups. No other factor is consistent across subgroups. Any additional factors, if included, explain only a single variable. Thus, we conclude that the three-factor model captures all robust factors. We also observe that the complexity of architecture, higher #feats, early epoch phases, and harder tasks are generally associated with lower PVE of **Dif-gt**. We conjecture that when the decision boundary of the data or the model is complex, or there is misalignment between the two, **Dif-gt** wields less explanatory power. However, trends in secondary/tertiary factors and unexplained variance are inconsistent, and current data are insufficient for definite conclusions.

### 3.5 HOW ARE EARLY-TRAINING AND LATE-TRAINING DIFFICULTY DIFFERENT? (RQ5)

Epoch-wise subgroup analysis (Figure 8) shows that early in training, there is more variance not explained by **Dif-gt** compared to later in training. The extended correlation matrix in Figure 9 (Ap-

pendix 5.5) displays a similar trend where early-training metrics tend to have lower autocorrelation. They also correlate more weakly with middle- or late-phase difficulty measurements. This suggests that early-epoch snapshots may be unnecessary for understanding model behavior at convergence.

## 4 DISCUSSION

**Factors and their Interpretation.** We consistently extract three orthogonal factors: **Dif-gt**, which captures ground-truth label-aware metrics (e.g., loss, AUM, and forgetting events); **Dif-conf**, which mainly captures label-agnostic metrics (confidence and ensemble agreement); and **Inf/Val**, which encompasses attribution-related quantities (influence, valuation, gradient similarity). We predict that other label-aware, label-agnostic, and influence/valuation metrics not included in our study load to the three factors, respectively. Metrics that primarily load onto one factor can inconsistently or weakly load onto other factors. The precise relationship between difficulty and attribution, in particular, remains challenging to describe.

**Comparison to Prior Works' Conclusions.** **Dif-gt** corresponds to the common factors of difficulty and outliers proposed by Kwok et al. (2024) and Carlini et al. (2019). We attribute our additional dimensions to choices in statistical analysis methods (e.g. PFA w/ manual factor count search), the mechanical diversity of the metrics and architectures, and inclusion of hard tasks.

**Dif-gt** and **Dif-conf** form a V-shape distribution reminiscent of Baldock et al. (2021)'s notion of train vs test set difficulty. We believe that **Dif-conf** is related to "train inference depth" dimension and **Dif-gt** with "val inference depth." Data that "looks like another label" causes overconfident and incorrect predictions (high **Dif-conf**, high **Dif-gt**); these data will be harder to learn during training (high train inference depth), and would cause early incorrect predictions as validation data (low val inference depth). The remaining data form a linear axis where some samples are straightforwardly easy (high **Dif-conf**, low **Dif-gt**; low train inference depth, low val inference depth) and some are straightforwardly hard. This hypothesis should be tested in future work.

We observe low correlations between the metrics that load onto **Dif-gt** and **Inf/Val**, and no interpretable shape when the joint distribution is plotted. These observations contradict the claim of Paul et al. (2021) that more difficult data are more important for generalization, and might explain why the random pruning baseline remains strong compared to pruning by data difficulty Liu et al. (2022).

**Applications of the Three-Factor Model.** Based on the three-factor model, we suggest measuring more than one metric for downstream applications. The SOTA noisy label detection methods that use two dimensions Deng et al. (2024); Forouzesh & Thiran (2024) might benefit from a third dimension. Prior interpretability methods for tabular data, such as difficult slice mining techniques Sagadeeva & Boehm (2021); Chang et al. (2024), should be generalized to multiple dimensions.

**Cost-Efficiency Implications.** Our experiments also provide practical guidelines for measurement under resource constraints. Metrics such as AUM, confidence, and kNN-Shapley are both robust and computationally feasible when mean-aggregated across epochs and across just three random seeds. Early-training snapshots provide noisier signals and can be discarded. There are diminishing returns to aggregating more than two or three metrics per factor. Finally, substituting noisier metrics (e.g., binary accuracy Kwok et al. (2024), gradient similarity) with more stable equivalents (e.g., AUM, kNN-Shapley) can improve downstream applications at no additional cost.

**Limitations and future work.** Important questions remain open. Our analysis is exploratory in nature, with the inherent limitations of factor analysis as a technique. There are limitations on the diversity of tasks, architectures, and metrics included. We do not have a sufficient diversity of datasets to generalize the effect of dataset metafeatures on the PVE of each factor. Our conclusions have not yet been tested on non-neural baselines (e.g., decision trees, SVMs), second-order metrics, model-agnostic metrics, or other modalities. Further experiments are needed to test the generality of these factors beyond TDL. Further research is also needed to test whether our educated recommendations for downstream applications are effective in practice.

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

# 5 APPENDIX

## 5.1 LLM USAGE DISCLAIMER

Large language models were utilized to retrieve papers for background research, for proofreading, and to aid in coding.

## 5.2 HYPERPARAMETER TUNING

For each task-architecture pair, we performed hyperparameter tuning using the tree-structured Parzen estimator (TPE) algorithm Bergstra et al. (2011) implemented in Optuna Akiba et al. (2019). Each task-architecture pair was tuned for 50 iterations using 3-fold mean cross-validation accuracy as the objective. A single ancestral seed was used for reproducibility purposes, and Optuna's default values were used for other parameters.

All model architectures shared the following hyperparameters:

1. Total number of epochs: 10-100. We did not use early stopping.
2. Optimizer: SGD, Adam Kinga et al. (2015), or AdamW Loshchilov & Hutter (2019).
3. Batch size: 32-1024, logarithmic scale.
4. Learning rate scheduler: constant or step decay.
5. Initial learning rate: 1e-5 - 1e-2, logarithmic scale.
6. LR step size, if using step decay: 5-50.
7. LR step multiplier ($\gamma$), if using step decay: 0.1-0.95.
8. Categorical embedding dimension: 4-128, logarithmic scale.

The following hyperparameters were specific to the model architecture:

1. MLP: Depth, width, shape (constant, pyramid, inverse pyramid).
2. ResNet: Number of blocks, width.
3. TabTransformer: Hidden dimensions, number of attention heads, number of transformer layers, whether to use [CLS] token.
4. SAINT: Hidden dimensions, number of attention heads, number of transformer layers, whether to prepend column tokens, mixup strength, cutmix strength.

## 5.3 HARDWARE DETAILS

An NVIDIA RTX A6000 GPU was used to train the models and measure difficulty metrics. Due to GPU memory constraints, for some task-architecture-metric pairs, lower precision (16-bit) inference had to be used during measurement.

## 5.4 OPEN SOURCE LIBRARY

We release an open-source Python library, EASYDATDIF, which simplifies the task of measuring one or more data difficulty metrics from a stream of model snapshots. Measuring over 300 million independent data points of data difficulty values over 4400 models and 250k epochs was made possible by this library. It takes as input an iterable of (model snapshot, model name, epoch) tuples, alongside the names of metrics to measure. It autotunes the batch size to maximize throughput, as the optimal batch size can vary widely between different metrics. It uses on-GPU data structures and streaming algorithms where possible to reduce communication costs. Additional metrics, additional axes of GPU parallelism, and more sophisticated autotuning will be supported in the future.

## 5.5 ADDITIONAL FIGURES

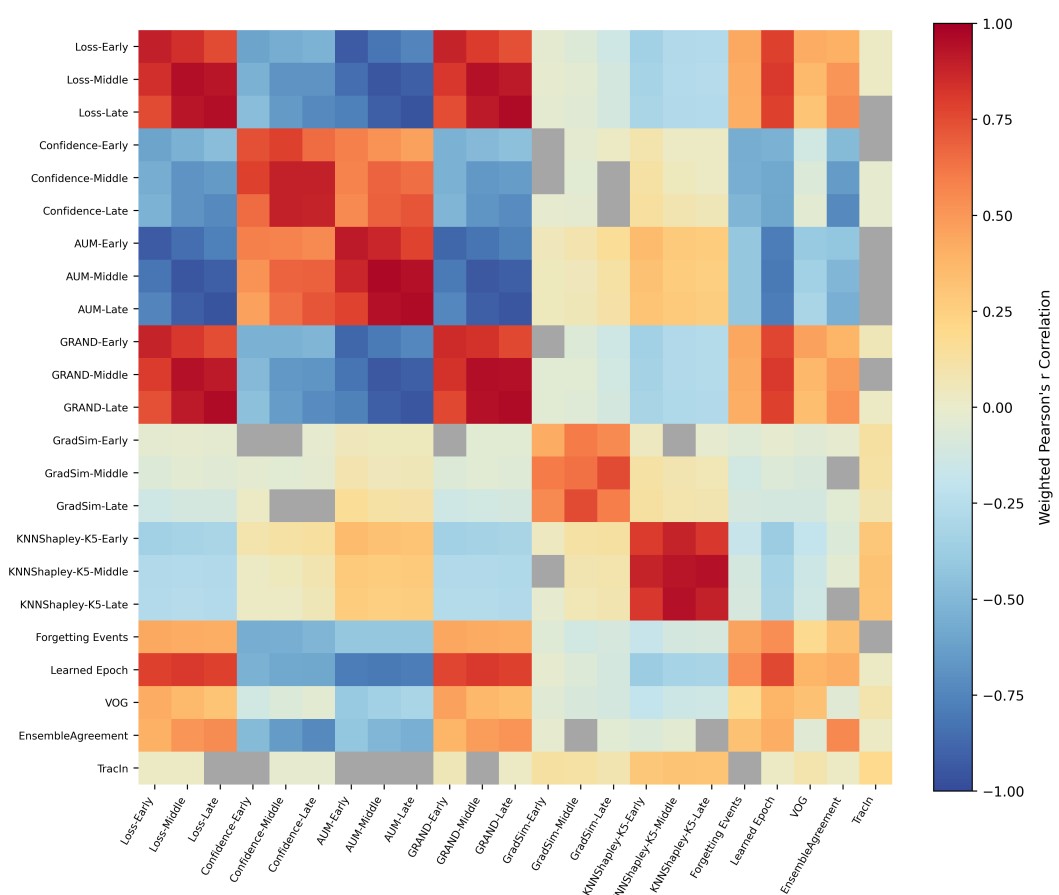

Figure 9: Correlation matrix between all metrics, where epoch-wise metrics have each been split into three separate metrics based on the phase of training: early, middle, and late.

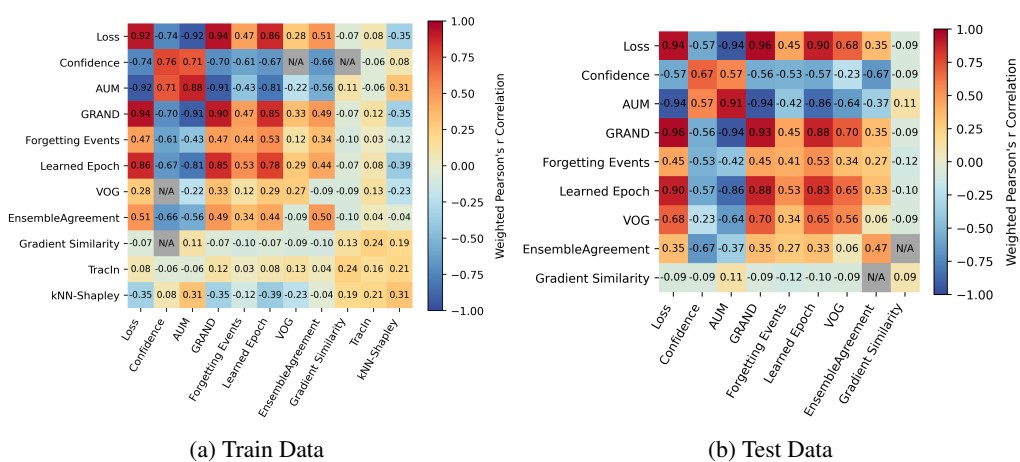

(a) Train Data

(b) Test Data

Figure 10: Correlation matrix between all metrics, split between train and test sets.

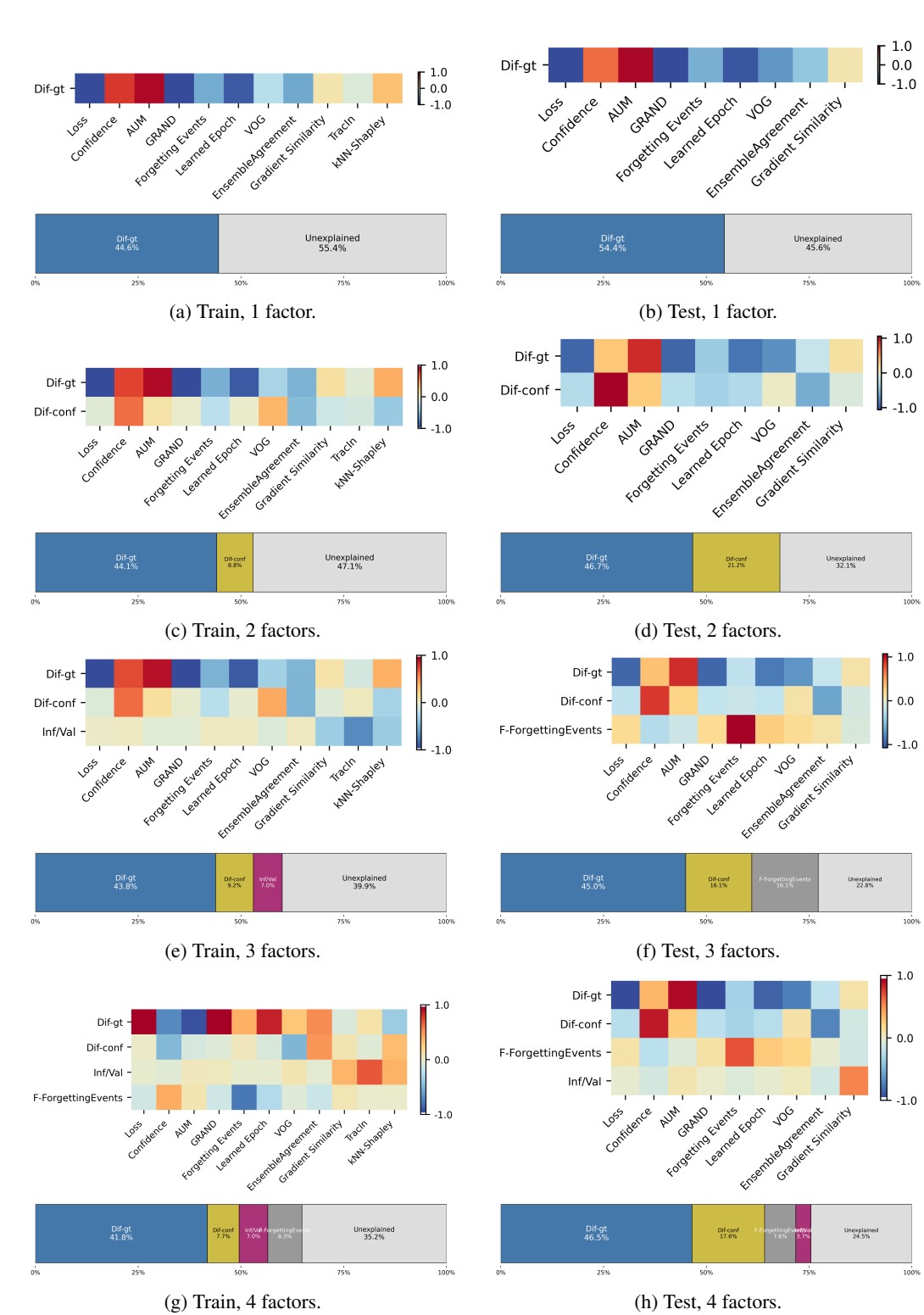

Figure 11: Factor analysis with varying number of factors by train/test split.

