# OpenReview forum: "What Data Difficulty Metrics Should We Measure for Tabular Deep Learning?"
_ICLR.cc/2026/Conference — ICLR 2026 Conference Withdrawn Submission_

### Official Review · Reviewer_zV5C · 2025-10-29

**Soundness:** 2
**Presentation:** 3
**Contribution:** 2
**Rating:** 4
**Confidence:** 4

**Summary:**

The paper conducts a systematic comparison of data difficulty/value measures for tabular deep learning. Across multiple datasets and architectures, it aggregates scores and applies factor analysis, arguing for a three-factor structure: label-aware difficulty, confidence, and influence/value. Based on these, it proposes low-cost metric combinations and practical guidance.

**Strengths:**

1. It focuses specifically on tabular data (like spreadsheets), which fills an important gap. Previous work often centered on images or videos, so this addresses a practical need that was overlooked.
2. The authors use straightforward statistical methods to combine many different performance measures into one clear summary. This makes the results reliable and easy to follow.
3. The explanation of the "three-factor" approach is simple and logical. Practitioners can quickly grasp how it works without needing advanced expertise.

**Weaknesses:**

1. The core work is an empirical synthesis of existing methods, with minimal advancement of theoretical foundations.
2. The choice of exactly three factors lacks objective justification. No analysis compares alternative models or quantifies sensitivity to factor count.

**Questions:**

1.Can you validate the proposed rankings with external objectives—e.g., curation guided by your “low-cost trio” improving generalization on unseen models/tasks?
2.How stable is the three-factor solution under alternative factor models and objective factor-number selection?
3.Do the factors persist (with similar semantics) when adding non-neural baselines and training-agnostic metrics?
4.Can you provide falsifiable predictions for the reported V-shapes and test them?
5.For high variance metrics, can you report cost variance trade-offs under different hyperparameters and snapshot sampling schemes?

---

### Official Review · Reviewer_Vyuf · 2025-11-01

**Soundness:** 3
**Presentation:** 2
**Contribution:** 2
**Rating:** 2
**Confidence:** 4

**Summary:**

This paper investigates the notion of data difficulty in tabular deep learning (TDL). The authors conduct a comprehensive empirical analysis of 11 representative difficulty metrics, covering logit-based, gradient-based, ensemble-based, and influence/valuation-based metrics. During the analysis, four model architectures (MLP, ResNet, TabTransformer, and SAINT) are employed. After experiments and statistical factor analysis, the authors identify three orthogonal factors underlying data difficulty:  Label-aware difficulty (Dif-gt), Confidence (Dif-conf), and Influence/Valuation (Inf/Val). The authors also conduct an analysis of the proposed three-factor model for common downstream applications of difficulty ranking and noisy label detection.

**Strengths:**

1. The topic of how to measure and interpret data difficulty for tabular deep learning is interesting.
2. The paper provides a comprehensive empirical analysis of data difficulty tabular deep learning, which covers 11 metrics, multiple architectures, and numerous datasets.
3. The authors found three orthogonal factors (Dif-gt, Dif-conf, and Inf/Val).

**Weaknesses:**

1. This paper does not provide a theoretical discussion on why these three orthogonal factors (Dif-gt, Dif-conf, and Inf/Val) naturally arise and how they relate to model generalization.
2. While the paper focuses on tabular DL, it would be valuable to test whether the proposed three-factor model can be generalized to other modalities (e.g., text, vision, or audio).
3. The paper employs principal factor analysis (PFA) with manual factor count selection.
4. The paper uses principal factor analysis (PFA) with manual selection of the number of factors. This process may involve subjective decisions. Can the authors analyze the sensitivity of the number of factors?

**Questions:**

Suggestions:

Some font sizes in Figures 2, 3, 4, and 8 are too small to read clearly. Can the authors enlarge the text?

---

### Official Review · Reviewer_hEHT · 2025-11-01

**Soundness:** 3
**Presentation:** 2
**Contribution:** 2
**Rating:** 4
**Confidence:** 3

**Summary:**

- The paper notes that while "data difficulty" metrics are widely used for tasks like noisy label detection and data pruning, they are often chosen in an "ad hoc manner". Furthermore, systematic evaluations have been limited to vision, and tabular deep learning (TDL) presents unique challenges.
- The authors conduct a "comprehensive empirical study" by collecting 11 representative difficulty metrics (including logit-based, gradient-based, ensemble, and valuation methods) across diverse TDL architectures.
- The results "contradict both the view that difficulty metrics are neither redundant nor hyper-specialized". Instead, the study identifies three consistent factors: Label-aware difficulty, Confidence, Influence/Valuation.

**Strengths:**

- The paper tackles a practical problem, noting that researchers often select data difficulty metrics in an "ad hoc manner".
- The authors conduct a comprehensive empirical study. This study spans 11 diverse metrics , multiple tabular datasets , and modern TDL architectures.
- The paper's key contribution is a three-factor model that organizes the ad hoc metrics into three consistent, orthogonal factors.
- The authors contribute an open-source Python library that streamlines the measurement of difficulty metrics.

**Weaknesses:**

- The paper explicitly excludes all "non-neural baselines (e.g., decision trees, SVMs)" , which is a significant omission given that tree-based models are often state-of-the-art and widely used for tabular data.
- Despite aiming for a "comprehensive" study, the authors excluded several entire classes of metrics. These include "prohibitively expensive" metrics like second-order methods , "training-free or model-agnostic metrics" like instance hardness , and "trivial variants" of other metrics.
- The paper's primary contribution---the "three-factor model"---relies on factor analysis, which the authors concede is "exploratory in nature" and has "inherent limitations". The number of factors was chosen using a heuristic (preventing single-variable factors), and the interpretation of these factors is post hoc.
- The findings for noisy label detection are based on a dataset where 10% of labels were "randomly corrupted". This uniform, random noise model is a simplification and may not represent real-world label noise, which is often feature-dependent or asymmetrical.

**Questions:**

None.

---

### Official Review · Reviewer_GKqu · 2025-11-01

**Soundness:** 3
**Presentation:** 3
**Contribution:** 2
**Rating:** 2
**Confidence:** 2

**Summary:**

The paper presents an extensive empirical study about existing difficulty metrics used in tabular data. By extensively evaluating multiple models on several datasets, each is trained with different random seeds and train-test split, the aim is to determine the "principal components" of difficult metrics presented in the literature. From that, one can investigate how many metrics should be use for evaluation. The paper also considers the robustness of those metrics under the label noise setting in which 10 percent of training samples have their labels corrupted. Throughout an extensive statistical analysis, the results are unclear, making these problems remain open.

**Strengths:**

The paper presents a systematic analysis to investigate the effect or influence of existing metrics about data difficulty in tabular data. The empirical evaluation is performed following a statistical approach with multiple models randomized at different seeds, trained on several datasets whose train-test split is also carefully considered. In addition, a small percentage of label noise is also added into those datasets to see how robust those metrics are.

**Weaknesses:**

Despite a good motivation, the contribution of the current study is limited at empirical evaluation only. In addition, the conclusion of the study is inconclusive, which further reduces the interest of readers.

Another weakness is that the paper simply focuses on the empirical evaluation. It could be stronger if it has some theoretical analysis to understand the overlapping between different metrics, then use empirical results to validate such hypothesis.

**Questions:**

**Minors**
- For citation referring to studies or papers, they should follow the format (authors, year). For example: (Kwok et al., 2024). The current citation format, such as ty Kwok et al. (2024), is incorrect and causes difficulty when reading the paper. This format refers to the authors of that paper, not the paper itself. If natbit is use, consider \citep{} and \citet{}.
- Some figures (such as Figs 4-8) are unreadable due to small font size when printing.

---

### Note · Authors · 2025-12-03

**Comment:**

We are grateful for the reviewers’ constructive criticism. We are withdrawing the manuscript and will integrate the feedback into subsequent work.

**Withdrawal Confirmation:**

I have read and agree with the venue's withdrawal policy on behalf of myself and my co-authors.